# A deep learning framework for the early detection of multi-retinal diseases

**Sara Ejaz[1], Raheel Baig[2], Zeeshan Ashraf[2]\*, Mrim M. Alnfiai[3], Mona Mohammed Alnahari[3], Reemiah Muneer Alotaibi[4]**

**1** Department of Information and Technology, University of Gujrat, Gujrat, Punjab, Pakistan, **2** Department of Computer Science, The University of Chenab, Gujrat, Punjab, Pakistan, **3** Department of Information Technology, College of Computers and Information Technology, Taif University, Taif, Saudi Arabia, **4** Information Technology Department, College of Computer and Information Sciences, Al-Imam Mohammad Ibn Saud Islamic University, Riyadh, Saudi Arabia

\* zeeshan@cs.uchenab.edu.pk

**Data Availability Statement:** Data is available and can be provided without any restrictions. Dataset that has been used in this research is available via Kaggle at the following URL: https://www.kaggle.

## Abstract

Retinal images play a pivotal contribution to the diagnosis of various ocular conditions by ophthalmologists. Extensive research was conducted to enable early detection and timely treatment using deep learning algorithms for retinal fundus images. Quick diagnosis and treatment planning can be facilitated by deep learning models' ability to process images rapidly and deliver outcomes instantly. Our research aims to provide a non-invasive method for early detection and timely eye disease treatment using a Convolutional Neural Network (CNN). We used a dataset Retinal Fundus Multi-disease Image Dataset (RFMiD), which contains various categories of fundus images representing different eye diseases, including Media Haze (MH), Optic Disc Cupping (ODC), Diabetic Retinopathy (DR), and healthy images (WNL). Several pre-processing techniques were applied to improve the model's performance, such as data augmentation, cropping, resizing, dataset splitting, converting images to arrays, and one-hot encoding. CNNs have extracted extract pertinent features from the input color fundus images. These extracted features are employed to make predictive diagnostic decisions. In this article three CNN models were used to perform experiments. The model's performance is assessed utilizing statistical metrics such as accuracy, F1 score, recall, and precision. Based on the results, the developed framework demonstrates promising performance with accuracy rates of up to 89.81% for validation and 88.72% for testing using 12-layer CNN after Data Augmentation. The accuracy rate obtained from 20-layer CNN is 90.34% for validation and 89.59% for testing with Augmented data. The accuracy obtained from 20-layer CNN is greater but this model shows overfitting. These accuracy rates suggested that the deep learning model has learned to distinguish between different eye disease categories and healthy images effectively. This study's contribution lies in providing a reliable and efficient diagnostic system for the simultaneous detection of multiple eye diseases through the analysis of color fundus images.

com/datasets/andrewmvd/retinal-disease-classification.

**Funding:** This research was funded by Taif University, Saudi Arabia, Project No. (TU-DSPP-2024-41).

**Competing interests:** The authors have declared that no competing interests exist.

# 1 Introduction

The retina is a delicate layer located on the internal aspect of the human ocular organ. The major cause that people lose their eyesight or blurriness is due to age and some retinal diseases. Early detection of these diseases and proper diagnosis may prevent permanent vision loss. With appropriate treatment and consistent monitoring, it is feasible to decelerate or hinder additional deterioration of vision, particularly when the condition is identified during its initial phases [1]. Some causes of the damage in the retina are old age trauma and light damage. Some other diseases like diabetes, hypertension, and cholesterol may also affect the retina. Diabetic Retinopathy (DR), Macular Degeneration, Retinal Vein Occlusion (RVO), and Hypertensive Retinopathy cause damage to retinal vessels. Glaucoma is present when the optic nerve is damaged. When we get older macular holes happen. The effect of the macula hole is a blurred and not clear image.

Retinal diseases, such as DR, Age-Related Molecular Degeneration (ARMD), and glaucoma, are major contributors to blindness on a global scale. Timely identification and precise recovery from these conditions are essential for prompt treatment and the prevention of vision loss. However, identifying and classifying retinal diseases accurately and efficiently can be challenging for human specialists due to the complexity and variety of retinal images. Therefore, the development of an automated retinal disease classification system using deep learning or neural network models can significantly enhance the precision and speed the detection and treatment. Glaucoma is a group of ocular conditions leading to harm to the optic nerve. Internationally, the primary factors contributing to vision impairment include [2]:

- ARMD

- Cataract

- DR

- Glaucoma

- Uncorrected refractive errors

To detect retina disease, various medical tests like Fundus photography, Optical Coherence Tomography (OCT), and Fluorescein angiography are performed. A retinal camera, also referred to as a fundus camera, is a specialized instrument that integrates a microscope with low power and a built-in camera. Its purpose is to capture detailed photographs of the eye's internal structures, such as the retinal layers, vascular network, optic nerve head, macular region, and posterior segment. By utilizing this technology, healthcare professionals can obtain high-resolution images that aid in the recovery and ongoing monitoring of different ocular disorders [3]. OCT does not provide direct visualization of blood in the retina [4], so it may not be the optimal imaging modality for documenting or measuring diseases involving bleeding in the retina. As OCT primarily relies on measuring reflected light to create detailed cross-sectional images of the retina, it may not accurately capture the presence or extent of blood. In cases where bleeding or hemorrhage is suspected, fundus photography can be more effective in documenting the condition. Fundus photography captures a high-resolution image of the posterior eye, capturing the retina and vascular network. Non-invasive methods for early detection and cure of retinal diseases are essential to intercept or control vision loss. Fundus images, captured using monocular cameras, provide a non-invasive and cost-effective technique for large-scale screening of fundus diseases. Fundus image-based eye diagnosis relies on various biomarkers, including optic cup, optic disc, blood vessels, fovea, macula, and specific lesions like hard exudates, hemorrhages, and microaneurysms used in DR diagnosis.

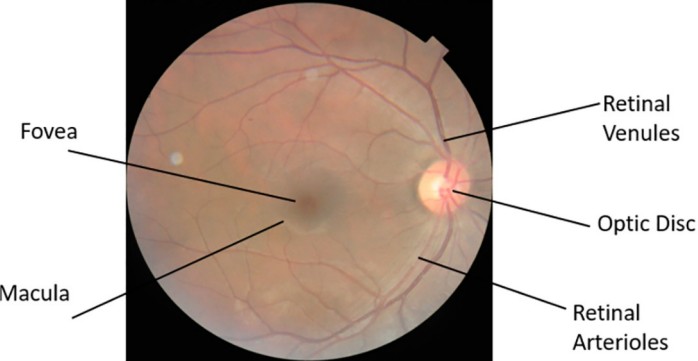

**Fig 1. Human eye.**

Diabetes patients constitute a significant portion of the population with eye-related issues. DR, the most common diabetic eye condition, often lacks early symptoms but poses a significant risk of blindness and is among the top four causes of blindness. Early detection of DR is crucial for successful treatment and to avoid poor visual outcomes. Media Haze (MH) is a key indicator of cataracts, a widespread eye disease. Detecting MH in its early stages is essential for early healthcare to reduce the risk of sight deprivation associated with cataracts. ARMD, linked to aging, affects central vision, leading to visual impairment. Optic Disc Cupping (ODC) is frequently associated with glaucoma and other eye conditions, resulting from reduced Ocular nerve blood circulation or increased pressure in the fundus. Timely treatment is often lacking, causing rapid vision decline and severe impairment. Fig 1 illustrates the structure of the human eye.

## 1.1 Research objective

The following are the research objectives of the suggested approach:

- Create a Deep Learning (DL) model designed for the multi-class classification of retinal images.

- Achieve high accuracy in the automated detection of common eye disorders, involving DR, MH, and ODC.

- Assess the model's performance on massive and wide-ranging datasets to ensure generalizability and reliability.

- Investigate the potential integration of the developed model into existing healthcare systems for seamless adoption by eye care professionals.

- Explore the model's contribution to early disease detection, with a focus on improving patient outcomes and minimizing vision loss.

- Evaluate the scalability and efficiency of the proposed solution for widespread use, particularly in regions with limited access to healthcare resources.

- Examine the interpretability of the deep learning model to enhance trust and understanding among healthcare practitioners.

## 1.2 Research contribution

Here are Key contributions of the suggested methodology:

1. The retina diseases like DR, MH, and ODC are identified at an initial phase is crucial to avert irreversible vision impairment.

2. In the field of biomedical research, extensive evidence supports the superiority of deep convolutional networks that have undergone pre-training on massive datasets, compared to deep models trained from scratch.

3. The experiments utilize the publicly accessible RFMiD and RFMiD 2.0 datasets. To mitigate the consequences of limited datasets data augmentation techniques are applied. Distinct researches are conducted using both augmented and initial datasets to compare performance.

The rest of the paper is organized as follows. Section 2 presents an overview of existing literature. In Section 3, we present materials and methodology. Section 4 presents results. Section 5 demonstrates comparisons between the results. Finally, Section 6 concludes the paper.

## 2 Overview of existing literature

In the medical field computer-assisted diagnosis is used for the diagnosis of diseases at their initial stages and to avoid permanent loss. Disease classifications are used to classify diseases in many medical fields.

The issue that is faced by ophthalmologists for computer-aided diagnosis is the limited number of datasets. In 2021 after seeing the vision loss rate which reaches 2.2 million [5]. Researchers have discovered that over 7 million individuals worldwide are currently experiencing irreversible vision impairment, with more than 1 million of them being Americans affected by total blindness [6]. Pachade, S published a dataset RFMiD with 3200 fundus images that contain 45 conditions of retinal disease [7]. RFMiD is the only dataset that includes a large number of diseases that appear in a clinical setting.

Almustafa et al. use the STARE [8] dataset to classify 14 ophthalmological defects using algorithms ResNet-50, EfficientNet, InceptionV2, 3-Layers CNN, and Visual Geometry Group (VGG). They concluded that EfficientNet gives the best accuracy at 98.43% [9].

Choudhary et al. use the dataset [10] to classify three retinal diseases and normal images of the retina. The model comprises 19 layers of CNN and obtained an accuracy of 99.17% with 0.99 sensitivity and 0.995 specificities [11].

Sengar et al. extract multi-class images from multi-label datasets RFMiD [7]. They classify the disease DR, MH, ODC, and normal images. To increase the extent of the dataset they formed a data transformation technique and compared the results of the proposed EyeDeep-Net algorithm with other algorithms VGG-16, VGG-19, AlexNet, Inception-v4, ResNet-50, and Vision Transformer. The obtained accuracy for validation is 82.13% and for testing 76.04% [12].

Pan et al. proposed a model in which they classify macular degeneration, tessellated, and normal retina. Their aim is to early recognition and treatment for retinal diseases. They used fundus images collected from China's hospital and applied deep learning models Inception V3 and ResNet-50. After adjusting hyperparameters and fine-tuning them according to their classifier they attained an accuracy rate of 93.81% from ResNet-50 91.76% when utilizing Inception V3 [13].

Kumar & Singh collects data from different datasets that are Messidor-2 [14], EyePACS [15], ARIA, and STARE [8] and classifies into 10 groups. They classify different stages of

diabetic retinopathy and Normal Fundus images. The proposed methodology consists of pre-processing, and a match filter approach, and for segmentation and classification post-processing steps are included. The model generates results based on accuracy, precision, recall, and F1-score measure that's 99.71%,98.63% 98.25% and 99.22% respectively [16].

[17] used a DL approach to capture the features and Machine Learning (ML) algorithms to classify glaucoma. The experiments are performed for the DRISTHI-GS [18] and ORIGA [19] dataset using 101 images and obtain a maximum training accuracy of 1.000.

Pandey et al. aimed to classify multiple retinal diseases. They classify glaucoma, AMD, DR, and healthy retinal images. They used DiaretDB [20], Drishti-GS [18], DRIVE [21], HRF [22], IDRiD, Kaggle-39 [23], Kaggle-DR, ODIR [24], MESSDIDOR [25], ORIGA-light [19], REFUGE [26], and STARE [8] datasets. InceptionV3 model of CNN is used, and the ImageNet dataset is used for initial weights pertaining. They classify three diseases DR, Glaucoma, AMD, and one class for healthy images [27].

The author [28] suggests a framework that is used for multi-disease comprises a combination of neural architectures in an ensemble configuration. First, they perform preprocessing steps by normalizing, image enhancement, and resizing. Then he detects the presence of diseases in the fundus image and performs multi-class classification. For disease risk detection convolutional neural networks that is Densenet201 and EfficientNetB4 were used. For disease classification, ResNet105 is added. RFMiD [7] is utilized for training. and validation. ODIR [19] dataset is applied in the testing phase. They classify 27 diseases.

Ho et al. use RFMiD [7] Data that contain fundus images. They selected five CNN architectures that trained to anticipate the existence of disease and classify the 28 abnormalities [29].

Abbas et al. also perform multi-class classification. He conducted tests on the 27 primary classes within the RFMiD dataset. He scored an area under curve (AUC) of 0.973. Their model selection is lighter. They use EfficientNetB4 and EfficientNetV2S for classification [30].

[31] performed augmentation techniques because their dataset contains only 69 images depicting vascular diseases, along with 55 healthy images. They use 10 epochs to train the multilayer deep CNN. With 10 epochs accuracy is 88.4%.

[32] introduces a compact convolutional neural network for automatic DR detection using four retinal image datasets. Utilizing 12-fold cross-validation, our model achieved high accuracy: 79.96% on the Diabetic Retinopathy Detection dataset, 94.75% on Messidor-2, 96.74% on IDRiD, and 89.10% on RFMiD, demonstrating its effectiveness across various datasets and enhancing ophthalmic diagnostics.

The author [33] proposed different models to classify vein occlusion disease and healthy class. For healthy images, the specificity is 100% and sensitivity, F1 score, and an accuracy 95%, 97%, and 97% respectively. They also compare specificity sensitivity F1 score and accuracy on ResNet18, ResNet18+SE, ResNet18+CBAM, and ResNet18+CA algorithms. [34] also used pre-trained models for retinal disease classification.

## 3 Materials and methods

In this article, we proposed a DL Technique for identifying retinal disorders through fundus images. Data was gathered from two datasets RFMiD [7] and RFMiD 2.0 [35]. The images in these datasets were single as well as multi-labeled. We separated single-label diseases and selected the diseases with more images in the dataset. We selected four classes. After acquiring the dataset we performed pre-processing steps which are shown in Fig 2. In preprocessing, we employed data augmentation to expand and balance the dataset, crop the unwanted area then resize the images to the same size because the images in the dataset were in different sizes. We partitioned the dataset into training and testing subsets. We converted Images in an array to

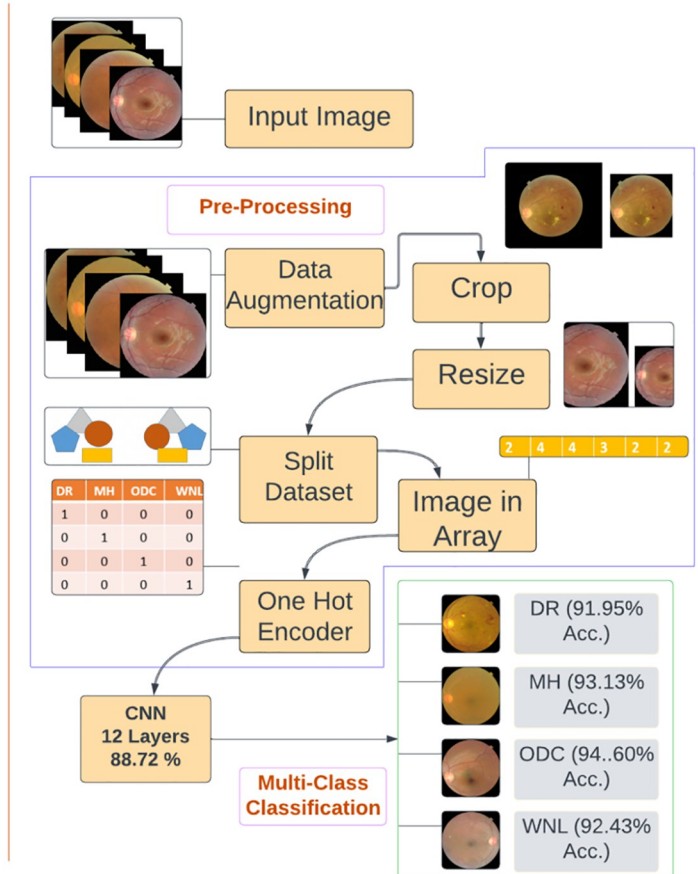

**Fig 2. Pre-processing.**

reduce the computing time and perform one hot encoder. Further, we implemented three CNN models to classify three retinal diseases and one healthy class. Firstly, the model was trained with the original dataset. To increase model performance and reduce overfitting, experiments were performed again to measure the results after data augmentation. The statistical results for augmented data were mentioned in the context of accuracy, specificity, sensitivity, precision, recall, F1 Score, and support. Graphically results are shown in terms of accuracy, loss, and confusion matrix.

## 3.1 Data gathering

This article's data was collected from public repositories, RFMiD [7] and RFMiD 2.0 [35]. The problem of detecting multiple eye diseases simultaneously was simplified by transforming it into a multi-class classification problem. Each image was assigned to a single disease class rather than having multiple labels. Unique images that exclusively belong to a single disease class were considered to ensure effective training of the neural networks. While recognizing that a retinal image could potentially exhibit multiple diseases, the decision to adopt a multi-class classification approach was driven by the need for simplicity, model training efficiency, dataset balance, label quality, and specific diagnostic goals. This approach ensures that the neural networks are effectively trained and evaluated, providing reliable and interpretable results

(a) (b) (c) (d)

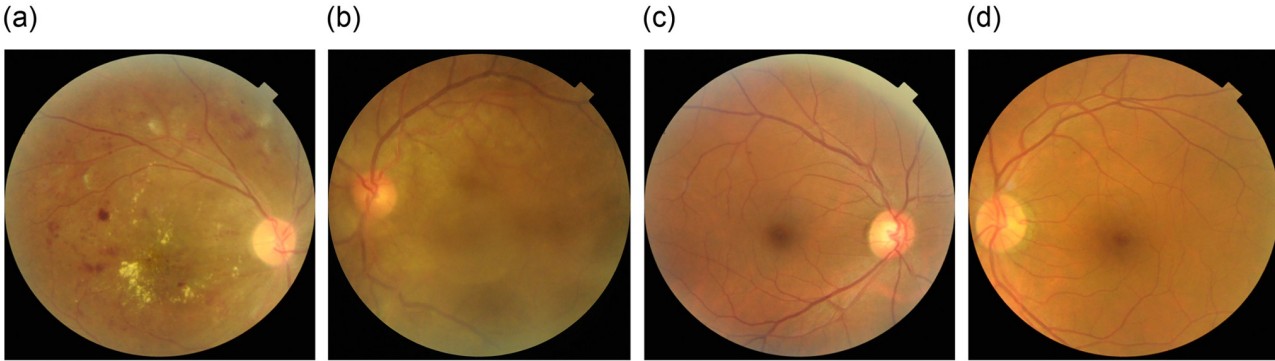

**Fig 3.** (a) Diabetic Retinopathy (b) Media Haze (c) Optic disc cupping (d) Normal.

that are immediately applicable in clinical settings. By focusing on unique images in each class, the dataset was appropriately balanced, allowing for accurate training and evaluation of the neural networks. For the final dataset preparation, we have chosen a total of four classes. Among these classes, one represented the normal (WNL) category, while the remaining three classes were related to different diseases. These diseases include DR, MH, and ODC as shown in Fig 3. By including these specific classes in the dataset, we aimed to capture a range of conditions related to eye health and provide a comprehensive representation of both healthy and diseased states. Table 1 shows the overall quantity of images which is single-labeled in both datasets.

## 3.2 Pre-processing

Pre-processing is the process of improving and enhancing image quality visualization. This was likely one of the pivotal factors influencing the success and accuracy of the subsequent stage in the proposed method. Medical images might contain additional content a problem that could cause poor image visualization. Poor-quality images can lead to unsatisfactory results. In the pre-processing stage, we performed data augmentation, cropping, resizing, dataset splitting, images in arrays, and a one-hot encoder to improve model efficiency.

**3.2.1 Data augmentation.** To improve the dataset and enhance the model's capacity for image handling from different perspectives, image augmentation techniques were employed as authors [12, 36, 37] used. These techniques significantly augmented the dataset size and helped capture the diverse variations of fundus images encountered in real-world conditions. The selection of augmentation methods was based on the understanding that fundus images can exhibit various transformations. The selected extension methods included various geometric transformations, such as rotations of 15˚, 30˚, and 45˚, and horizontal flips as [12] applied for fundus images. By applying these augmentation techniques, the dataset was enriched with variations of the original sample image. This augmentation process expands the dataset's diversity

**Table 1. Data distribution.**

| Classes | RFMiD | RFMiD 2.0 | Total |
|---------|-------|-----------|-------|
| DR | 401 | 70 | 471 |
| MH | 315 | 19 | 334 |
| ODC | 155 | 17 | 172 |
| WNL | 669 | 262 | 931 |

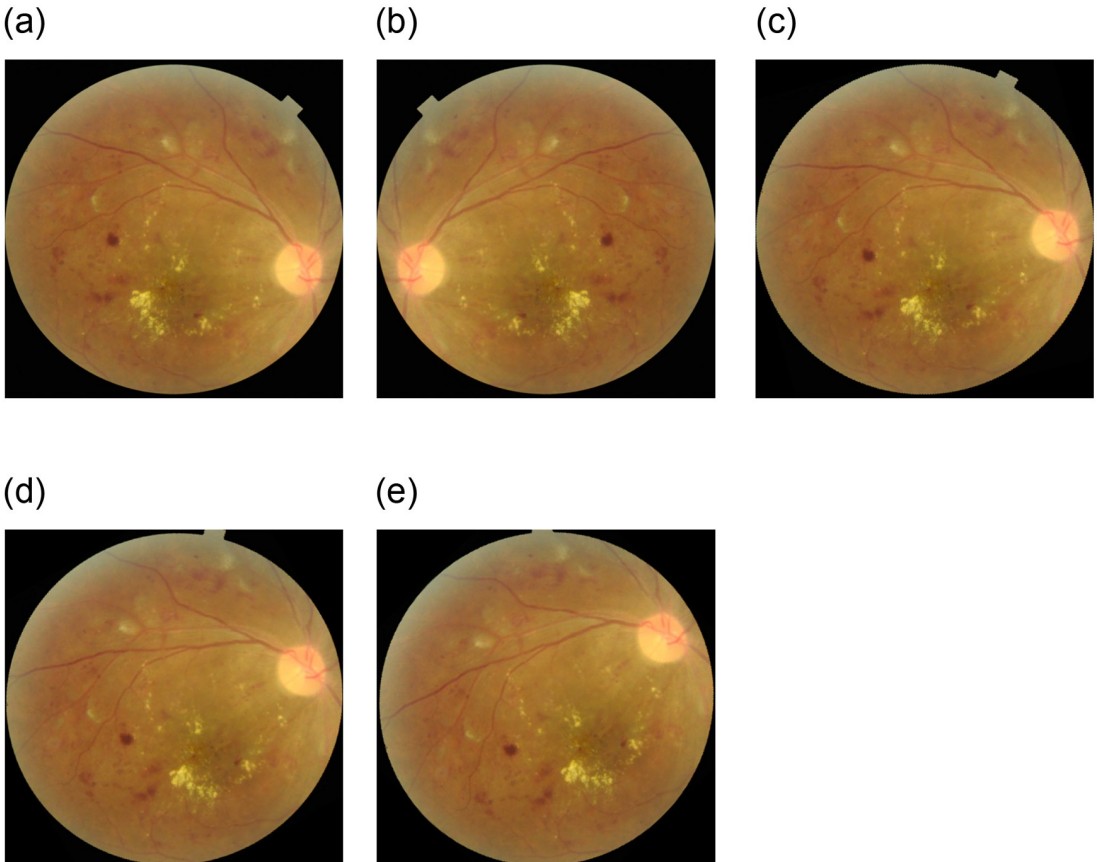

**Fig 4.** (a) Original (b) Flipped (c) 15˚ Rotation (d) 30˚ Rotation (e) 45˚ Rotation.

and enables the model to learn from a broader range of image variations, leading to improved performance and robustness. Fig 4 offers a visual depiction of the various image variations of DR i.e. Horitenzatal flip, rotation 15˚, 30˚, and 45˚ respectively, obtained after applying the augmentation techniques to the original sample image. Table 2 represents data for all classes before and after augmentation.

**3.2.2 Crop.** Cropping for feature extraction is a common technique used in image processing and computer vision tasks. By cropping, we reduce the amount of data that needs to be processed. This can significantly speed up the feature extraction process, especially when dealing with large images or datasets.

**3.2.3 Resize.** Resizing images is an important preprocessing step in computer vision, particularly in deep learning. One of the reasons for resizing images is to accelerate the training

**Table 2. Data distribution.**

| Diseases Name | Before Augmentation | After Augmentation |
|---|---|---|
| DR | 471 | 2361 |
| MH | 334 | 2367 |
| ODC | 172 | 2354 |
| WNL | 931 | 2360 |

process. When working with larger input images, DL models need to process a larger number of pixels, which significantly increases the complexity of computation and training duration.

By decreasing the size of images, the number of pixels that the model needs to learn from is reduced. This reduction in input size leads to a decrease in computational requirements, resulting in faster training. Training on smaller images allowed for quicker iterations and experimentation, making the development process more efficient [38]. Images in datasets are in different dimensions such as 2144 x 1424 x 3, 4288 x 2848 x3, and 512 x 512 x 3. We resized the image to 224 x 224 x 3 to reduce computational requirements and allow for quicker iterations and experimentation, making the development process more efficient.

**3.2.4 Split dataset.** In ML and data analysis, distributing the data into training and testing categories is a common practice. The main reason for this is to analyze performance metrics and model the generalization capability of an ML model. By splitting the dataset into training and test samples, we can ensure that the model undergoes training and evaluation in a robust and unbiased manner, enabling us to make informed decisions about its performance and generalization capabilities. In this experiment, 70% of the dataset was partitioned for training, 20% for testing, and 10% for validation.

**3.2.5 Image in array.** Converting an image into an array is a common practice in image processing and computer vision tasks. This conversion allows images to be manipulated, analyzed, and processed using mathematical and algorithmic techniques. Many computer vision algorithms involve extracting features such as edges, corners, or textures from images. This process is more straightforward when the image is represented as an array.

**3.2.6 One-hot encoder.** One-hot encoding is a widely practiced approach in DL to represent categorical variables as binary vectors. This method transforms categorical data into a numerical format, facilitating its processing by machine learning algorithms, including deep learning models.

## 3.3 Proposed deep learning architecture

Three deep learning architectures were proposed in this article and the results were examined with the original dataset as well as with the augmented dataset. The selection of CNN architectures with 12, 14, and 20 layers was a strategic decision to explore the trade-offs between model complexity, feature extraction capabilities, and computational efficiency. The 12-layer CNN was highlighted as the proposed methodology due to its high accuracy, balanced training time, and reduced risk of overfitting. The 14-layer CNN, while offering deeper feature extraction, did not outperform the 12-layer model. The 20-layer CNN, despite achieving high accuracy, showed signs of overfitting, indicating that a more complex model is not necessarily better for this specific task.

**3.3.1 Deep CNN-1 architecture.** Classification is a critical step in distinguishing between diseased and healthy retinal images. For image classification, we use different CNN layers. The sequence of the layers is given in Table 3. Convolutional layers are fundamental components of CNNs because they are designed to exploit the spatial structure of data, capture local patterns, share parameters to reduce redundancy and learn hierarchical representations. These properties make CNNs highly effective for tasks involving visual data, such as image classification.

**3.3.2 Feature extraction.** Feature extraction stands as a pivotal element within the model. A dedicated CNN model was trained for this purpose. The employed CNN model is constructed with a series of convolutional layers, including 2D convolutional layers, batch normalization layers, and 2D max pooling, along with dropout and dense layers. The introduction of filters facilitates the transfer of the dataset through each convolutional layer. Each

**Table 3. Model summary for CNNs.**

| CNN-1 | CNN-2 | CNN-3 |
|---|---|---|
| Convolutional Layer -1 | Convolutional Layer-1 | Convolutional Layer-1 |
| Max Pooling-1 | Max Pooling-1 | Batch Normalization-1 |
| Convolutional Layer-2 | Convolutional Layer-2 | Max Pooling-1 |
| Max Pooling-2 | Max Pooling-2 | Convolutional Layer-2 |
| Convolutional Layer-3 | Convolutional Layer-3 | Max Pooling-2 |
| Max Pooling-3 | Max Pooling-3 | Convolutional Layer-3 |
| Convolutional Layer-4 | Convolutional Layer-4 | Max Pooling-3 |
| Max Pooling-4 | Max Pooling-4 | Convolutional Layer-4 |
| Flatten-1 | Convolutional Layer-5 | Batch Normalization-2 |
| Dense-1 | Max Pooling-5 | Max Pooling-4 |
| Dropout-1 | Flatten | Convolutional Layer-5 |
| Dense-2 | Dense-1 | Max Pooling-5 |
| | Dense-2 | Convolutional Layer-6 |
| | Dense-3 | Batch Normalization-3 |
| | | Max Pooling-6 |
| | | Flatten |
| | | Dense-1 |
| | | Batch Normalization-4 |
| | | Dropout |
| | | Dense-3 |

convolutional layer extracts relevant information before the final max pooling. Finally, feature extraction is done through fully connected layers. The convolutional operation, denoted as (*), is a mathematical process that takes two functions (f, g) as inputs and yields a third function denoted as (f*g). In the context of image processing, convolution is carried out using a kernel, which is a small matrix typically of size k x k. The kernel should be odd since an odd number ensures better symmetry around the center and minimizes the possibility of aliasing.

The kernel is applied by sliding it over an image's pixels, generating feature maps. In a CNN, multiple filters are utilized in every convolutional layer to extract high-level features. If the input dimensions of a fundus image are (p x q), and n kernels with a window size of k x k are employed, the resulting image dimensions will be n x ((p − k + 1) x (q − k + 1)). The network creates meaningful feature representations from the data by capturing various aspects of the input image.

The given model architecture consists of several layers, including Convolutional, Max-Pooling2D, and Dense layers as shown in Fig 5. The output shape of each layer indicates the dimensions of the feature maps generated at each layer. The input shape of the images is specified and the images are expected to have three color channels (RGB). In the Convolutional Layers, the initial convolutional incorporates 32 filters sized 3x3 and employs the ReLU activation function. It takes the input image and applies 32 different filters to extract various features from the image. The second convolutional layer (Layer-2) is equipped with 64 filters sized 3x3 and employs the ReLU activation function, extracting more complex features from the input. Subsequently, the third convolutional layer (Layer-3) integrates 128 filters of size 3x3, utilizing the ReLU activation function to acquire even more abstract features from the preceding layers. The fourth convolutional layer (Layer-4) incorporates 256 filters of size 3x3 and applies the ReLU activation function, further enhancing the feature extraction process. Following each convolutional layer, a max pooling layer is added, featuring a 2x2

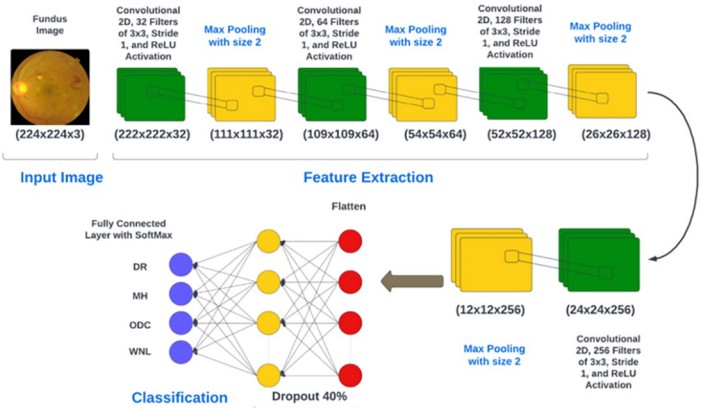

**Fig 5. CNN-1 architecture.**

pool size. This layer downsamples the output of the preceding convolutional layer by selecting the maximum value within each 2x2 region, aiding in reducing spatial dimensions while retaining essential features. After the final max pooling layer, a flattening layer is introduced to convert the 2D output into a 1D vector, Getting data ready for fully connected stages. The flattened output is then connected to a dense layer ('Dense') with 128 units and the ReLU function. This layer performs a linear transformation on the input data and introduces nonlinearity. To mitigate overfitting, a dropout layer is added with a dropout rate of 0.5. By dropping out some input elements, the network will not overdepend on specific features. Finally, the output layer is composed of the number of classes and uses the softmax activation. This produces probabilities for each class, determining the likelihood of the input image belonging to different classes. The model setup of an experiment is given in Table 4. The dataset contains nonlinearity, so the hidden layers in the CNN use a ReLU function. The final output layer utilizes the Softmax function. ReLU is a fast and efficient nonlinear activation function that outperforms alternatives like Sigmoid and Tanh, leading to quicker convergence. ReLU squashes negative activations in the feature map, enhancing accuracy and reducing training time according to Eq 1

$$\text{ReLU}(x) = \begin{cases} 0, & \text{if } x < 0, \\ x, & \text{if } x \geq 0. \end{cases} \tag{1}$$

**Table 4. Model setup for all CNNs.**

| Name | Parameter |
|---|---|
| Input | Fundus Images From Both dataset |
| Image size | 224 x 224 x 3 |
| Batch Size | 32 |
| Activation Function | Relu, Softmax |
| No of epochs | 20 |
| Dropout | 50% |
| Optimization Function | Adam Optimizer |
| Loss Function | Categorical Cross-entropy |
| L2 Regularization | 0.01 |

Softmax normalizes the network's output into probability scores. This enables the prediction of fundus image outcomes across four distinct classes: DR, MH, ODC, and WNL. Categorical Cross-Entropy (CCE) stands as one of the most prevalent loss functions employed in multi-class classification. It's used when the classes are mutually exclusive, meaning each input can belong to only one class. The predicted class probabilities are passed through a softmax activation, and the cross-entropy between the predicted probabilities and the ground truth labels is computed. The CCE loss is calculated as the negative log-likelihood of the true class probabilities given the predicted probabilities as given in Eq 2:

$$\text{CCE Loss} = -\sum_i t_i \cdot \log(y_i) \tag{2}$$

Where:

- $y_i$ represents the predicted probability for class $i$ (output of the softmax activation function) from the model.

- $t_i$ represents the one-hot encoded target label for class $i$. It's 1 if the true class is $i$ and 0 otherwise.

## 4 Results

We have conducted experiments to evaluate the proposed CNN model classification methodology, considering both qualitative and quantitative aspects. Our evaluation involved testing the proposed method using the data we collected.

### 4.1 Dataset

We have conducted experiments to evaluate the proposed CNN model classification methodology, considering both qualitative and quantitative aspects. Our evaluation involved testing the proposed method using the data we collected. We compiled a dataset comprising approximately 1908 images. We organized our dataset into four distinct classes, namely DR, MH, ODC, and Norma (WNL). At the outset, the dataset includes 334 images depicting MH, 471 images depicting DR, 172 images depicting ODC, and 931 images of WNL as shown in Table 1. After implementing data augmentation on the dataset to address the problem of data overfitting. Moreover, we encountered a significant class imbalance issue where the WNL class had a substantially higher number of images compared to the other classes. This created a challenge as it could potentially introduce biases in the results. To tackle this problem, we implemented data augmentation techniques to balance the classes. We got 2367 images of MH, 2261 images of DR, 2354 images of ODC, and 2360 images of WNL as shown in Table 2. Fig 6a is showing data distribution before Augmentation and Fig 6b for after augmentation.

For classification, we split the datasets into 70:20:10 for training, testing sets, and validation. This implies that 70% of randomly selected images were employed during the training phase, while 20% were set aside for testing and 10% was used for validation.

### 4.2 Experimental framework

In this study, trials were carried out on a 64-bit iteration of the Windows 10 operating system using Python. The system employed an Intel Core i5 7th Generation CPU, possessed 8 GB of RAM, and featured a storage capacity of 237 GB.

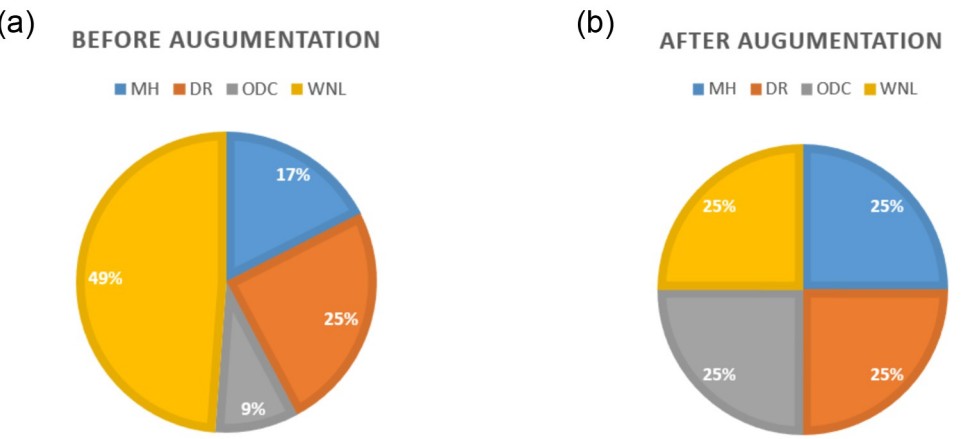

**Fig 6.** (a) Dataset Before Augmentation (b) Dataset After Augmentation.

### 4.3 Findings for feature extraction utilizing CNNs

In this section, feature extraction results are given in both statistical as well as graphical form. In statistical form, accuracy, specificity, sensitivity, precision, recall, F1 score, and support are given using the formula given in Eqs 3–8

$$\text{Sensitivity} \quad = \frac{TP}{TP + FN} \tag{3}$$

$$\text{Specificity} \quad = \frac{TN}{TN + FP} \tag{4}$$

$$\text{Accuracy} \quad = \frac{TP + TN}{TP + FP + TN + FN} \tag{5}$$

$$\text{Precision} \quad = \frac{TP}{TP + FP} \tag{6}$$

$$\text{Recall} \quad = \frac{TP}{TP + FN} \tag{7}$$

$$\text{F1-Score} \quad = 2 \cdot \left( \frac{\text{Precision} \cdot \text{Recall}}{\text{Precision} + \text{Recall}} \right) \tag{8}$$

**4.3.1 Results of feature extraction using CNN-1.** This section will delve into the results of feature extraction using CNN. The experiments employed the deep CNN base architecture model with training and testing data. The accuracy and loss charts for the suggested CNN model without data augmentation are presented in Fig 7a and 7b respectively. It is observable in the charts that the model initiates with a starting training accuracy of zero then gradually advancing with increasing epochs. The accuracy graph for CNN-1 without the data augmentation shows that there is overfitting in the model to reduce this we performed data augmentation.Table 5 presents the statistical results of Feature Extraction from the CNN-1 model without employing data augmentation. In the preliminary experiment without data

(a)
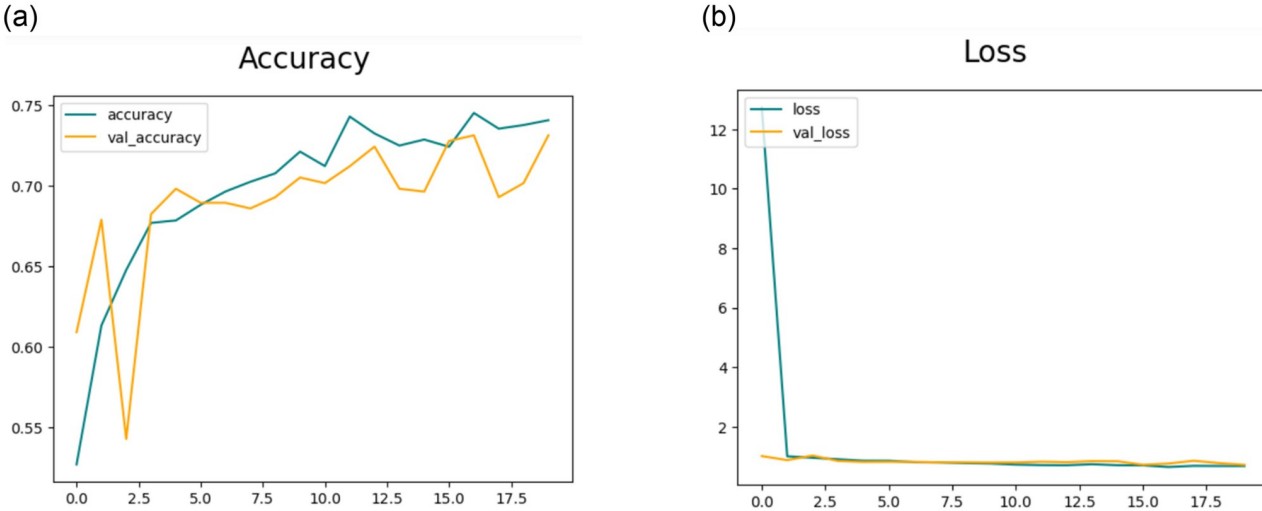

Fig 7. (a) Accuracy of the Model CNN-1 without the use of Data Augmentation (b) Loss of the Model CNN-1 without the use of Data Augmentation.

augmentation, the proposed model demonstrated accuracies of 83.94%, 90.39%, 90.39%, and 80.07% for DR, MH, ODC, and WNL, respectively.

Fig 8a illustrates the training and validation accuracy and Fig 8b presents loss for the CNN-1 model when utilizing augmented data. Conversely, in the second trial involving data augmentation, the proposed framework achieved 91.94%, 93.17%, 94.60%, and 92.43% accuracy rate for DR, MH, ODC, and WNL respectively as shown in Table 6. The experimental results indicate that the proposed architecture, when coupled with data augmentation, has achieved the highest accuracy.

The Confusion matrix for the CNN model is illustrated in Fig 9a in the absence of data augmentation, and Fig 9b depicting results for CNN-1 both with the inclusion of data augmentation.

**4.3.2 Results of feature extraction using CNN-2.** The section discusses the outcomes of feature extraction using CNN-2. Fig 10a depicts the training and validation accuracy and Fig 10b presents loss for the CNN-2 model when augmented data is not utilized.

Table 7 provides statistical results for the suggested CNN-2 model without data augmentation. In the preliminary experiment, the model attained accuracy rates of 82.90%, 87.26%, 90.05%, and 75.57% for DR, MH, ODC, and WNL, respectively.

Subsequently, in the second experiment with data augmentation Fig 11a depicts the training and validation accuracy and Fig 11b displays loss for the CNN-2 model when augmented data is utilized.

The model attained accuracy rates of 88.58%, 91.10%, 93.91%, and 90.12% for DR, MH, ODC, and WNL, respectively as shown in Table 8.

**Table 5. Class-specific statistics for the CNN-1 model with original data.**

| Classes | Accuracy % | Sensitivity % | Specificity % | Precision % | Recall % | F1 Score % | Support |
|---|---|---|---|---|---|---|---|
| DR | 83.95 | 83.29 | 86.03 | 61.58 | 86.18 | 71.88 | 136 |
| MH | 90.39 | 96.17 | 64.08 | 78.57 | 64.08 | 70.69 | 103 |
| ODC | 90.39 | 99.42 | 03.70 | 40.00 | 03.70 | 06.76 | 54 |
| WNL | 80.07 | 78.16 | 82.14 | 78.23 | 82.14 | 80.13 | 280 |

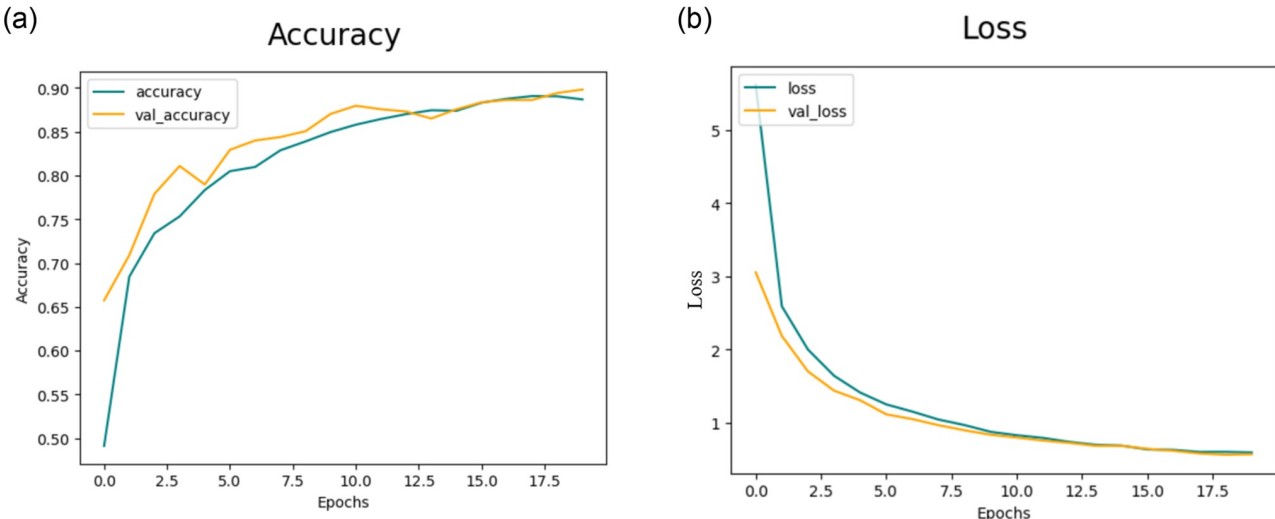

**Fig 8.** (a) Accuracy of the Model CNN-1 with the use of Data Augmentation (b) Loss of the Model CNN-1 with the use of Data Augmentation.

**Table 6. Class-specific statistics for the CNN-1 model with augmented data.**

| Classes | Accuracy % | Sensitivity % | Specificity % | Precision % | Recall % | F1 Score % | Support |
|---------|-----------|---------------|---------------|-------------|----------|------------|---------|
| DR | 91.94 | 98.20 | 92.35 | 77.26 | 95.96 | 85.65 | 470 |
| MH | 93.17 | 89.24 | 97.93 | 90.00 | 82.33 | 86.00 | 481 |
| ODC | 94.60 | 86.12 | 99.47 | 94.91 | 82.85 | 88.47 | 473 |
| WNL | 92.43 | 85.28 | 96.51 | 85.63 | 83.23 | 84.41 | 465 |

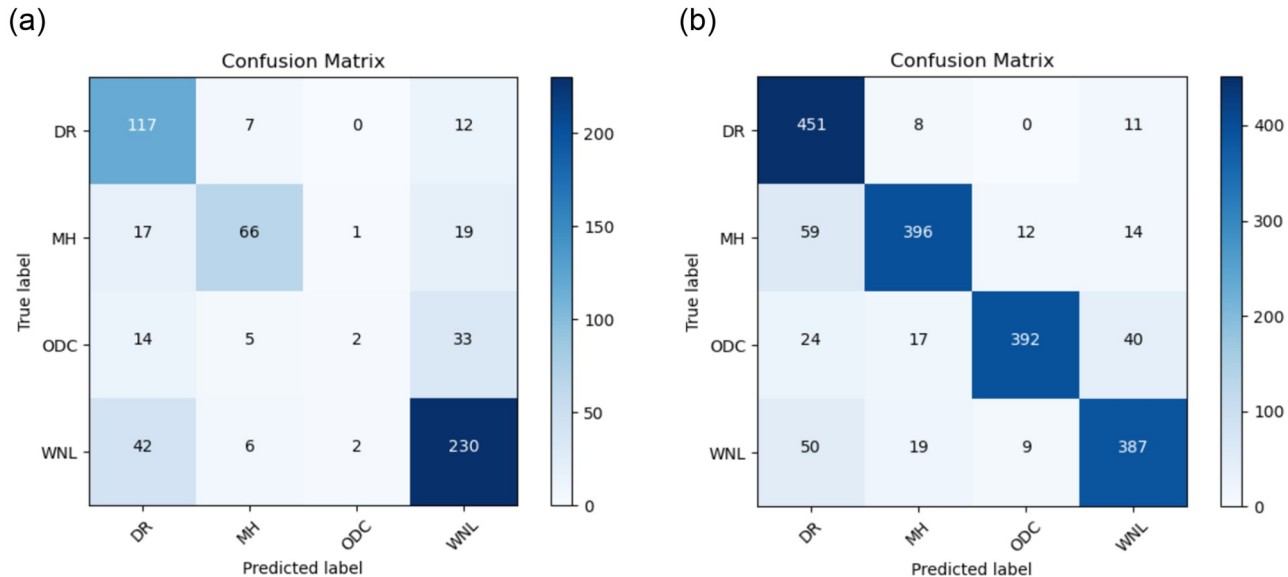

**Fig 9.** (a) Confusion Matrix for CNN-1 without the use of Data Augmentation (b) Confusion Matrix for CNN-1 with the use of Data Augmentation.

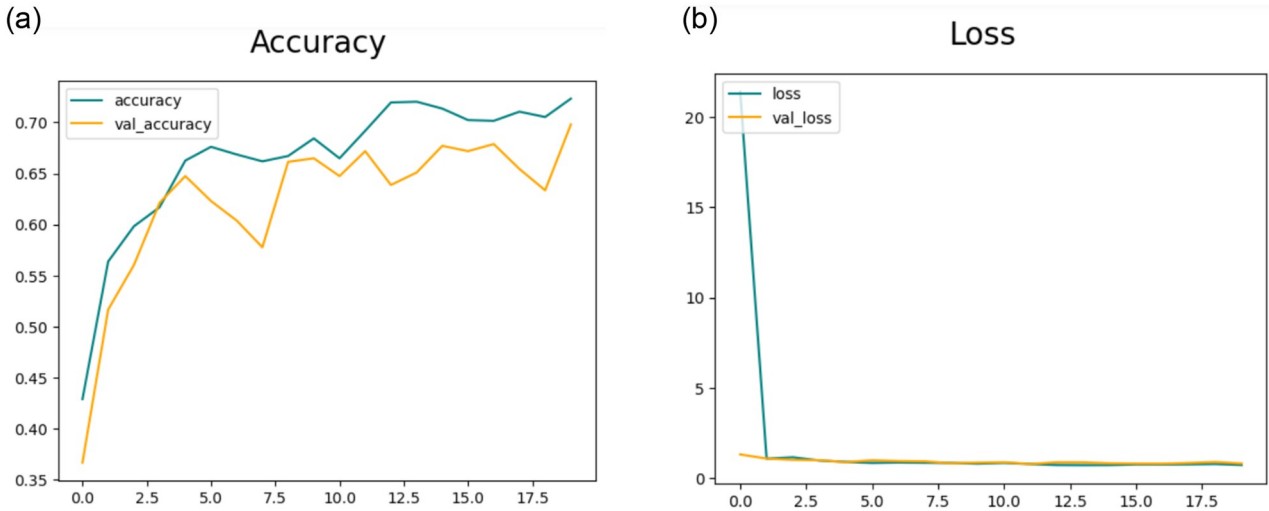

**Fig 10.** (a) Accuracy of the Model CNN-2 without the use of Data Augmentation (b) Loss of the Model CCN-2 without the use of Data Augmentation.

**Table 7. Class-specific statistics for the CNN-2 model with original data.**

| Classes | Accuracy % | Sensitivity % | Specificity % | Precision % | Recall % | F1 Score % | Support |
|---|---|---|---|---|---|---|---|
| DR | 82.90 | 86.49 | 84.78 | 60.67 | 79.41 | 68.83 | 136 |
| MH | 87.26 | 97.82 | 53.68 | 76.79 | 41.75 | 54.29 | 103 |
| ODC | 90.05 | 98.43 | 22.88 | 40.00 | 11.11 | 17.39 | 54 |
| WNL | 75.57 | 77.48 | 87.23 | 71.60 | 82.86 | 76.75 | 280 |

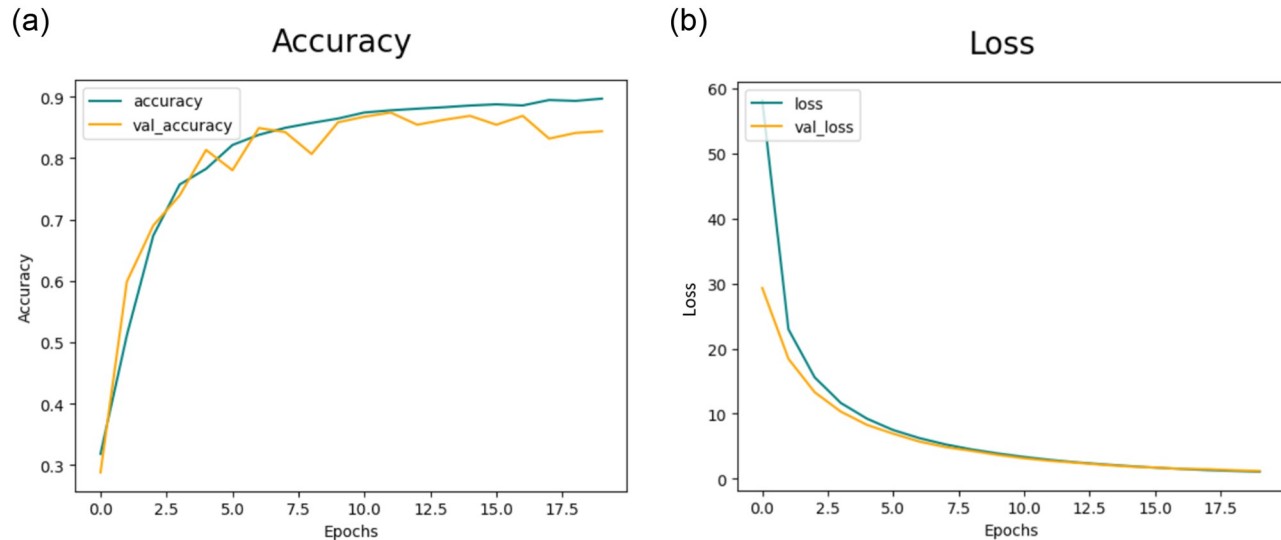

**Fig 11.** (a) Accuracy of the Model CNN-2 with the use of Data Augmentation (b) Loss of the Model CNN-2 with the use of Data Augmentation.

**Table 8. Class-specific statistics for the CNN-2 model with augmented data.**

| Classes | Accuracy % | Sensitivity % | Specificity % | Precision % | Recall % | F1 Score % | Support |
|---------|-----------|---------------|---------------|-------------|----------|------------|---------|
| DR | 88.58 | 97.83 | 86.89 | 69.67 | 95.32 | 80.42 | 470 |
| MH | 91.10 | 72.00 | 99.02 | 94.84 | 68.81 | 79.64 | 481 |
| ODC | 93.91 | 81.55 | 99.54 | 96.37 | 78.69 | 86.52 | 473 |
| WNL | 90.12 | 83.75 | 92.91 | 77.21 | 84.95 | 80.86 | 465 |

The Confusion matrix for the CNN-2 model is depicted in Fig 12a without the data Augmentation and Fig 12b shows with the data augmentation.

**4.3.3 Results of feature extraction using CNN-3.** The experiments employed the deep CNN base architecture model with both training and testing data. Fig 13a displays the accuracy and Fig 13b displays loss graph for the proposed CNN-3 model without utilizing data augmentation.

Table 9 provides the statistical results of the suggested CNN-3 model in the absence of data augmentation. In the initial trial without data augmentation, the Model CNN-3 achieved accuracy rates of 86.39%, 90.24%, 88.37%, and 82.30% for DR, MH, ODC, and WNL, respectively.

Subsequently, Fig 14a portrays the training and validation accuracy, and Fig 14b presents loss for the CNN-3 model when augmented data is employed.

Similarly, in the second experiment incorporating data augmentation, the proposed architecture attained accuracy rates of 93.90%, 95.51%, 96.20%, and 94.50% for DR, MH, ODC, and WNL, respectively, as detailed in Table 10.

The Confusion matrix for the CNN-3 model is presented in Fig 15a without data augmentation and in Fig 15b with data augmentation.

## 5 Comparisons

The purpose of this study is to evaluate the effectiveness of our model in detecting retinal diseases using the dataset provided. To assess how well the proposed CNN model distinguishes

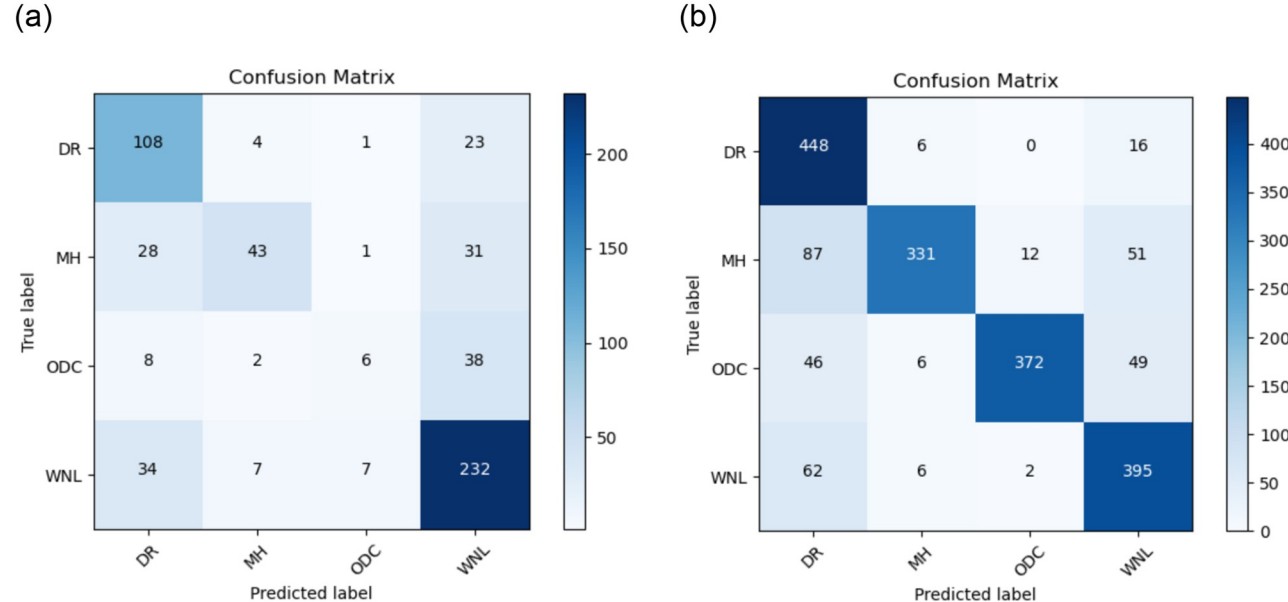

**Fig 12.** (a) Confusion Matrix for CNN-2 without use of Data Augmentation (b) Confusion Matrix for CNN-2 with use of Data Augmentation.

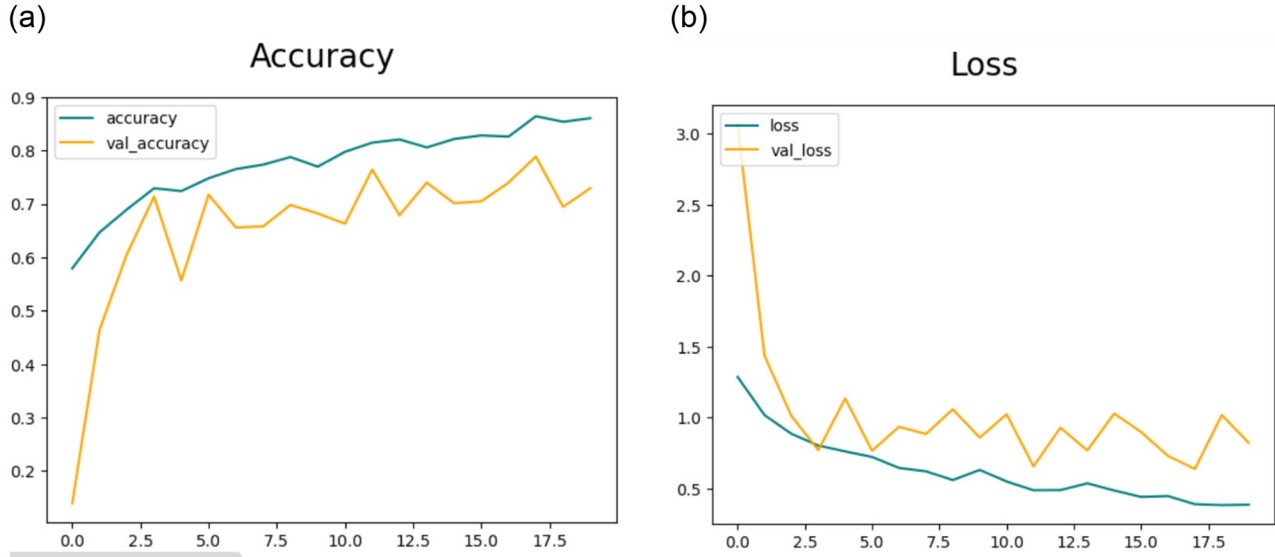

**Fig 13.** (a) Accuracy of the Model CNN-3 without Data Augmentation (b)Loss of the Model CNN-3 without Data Augmentation.

**Table 9. Class-specific statistics for the CNN-3 model with original data.**

| Classes | Accuracy % | Sensitivity % | Specificity % | Precision % | Recall % | F1 Score % | Support |
|---------|-----------|---------------|---------------|-------------|----------|------------|---------|
| DR | 86.39 | 86.95 | 84.55 | 66.86 | 84.56 | 74.67 | 136 |
| MH | 90.24 | 97.44 | 57.28 | 83.10 | 57.28 | 67.86 | 103 |
| ODC | 88.39 | 91.32 | 59.25 | 41.56 | 59.26 | 48.78 | 54 |
| WNL | 82.30 | 87.37 | 77.14 | 85.30 | 77.14 | 81.04 | 280 |

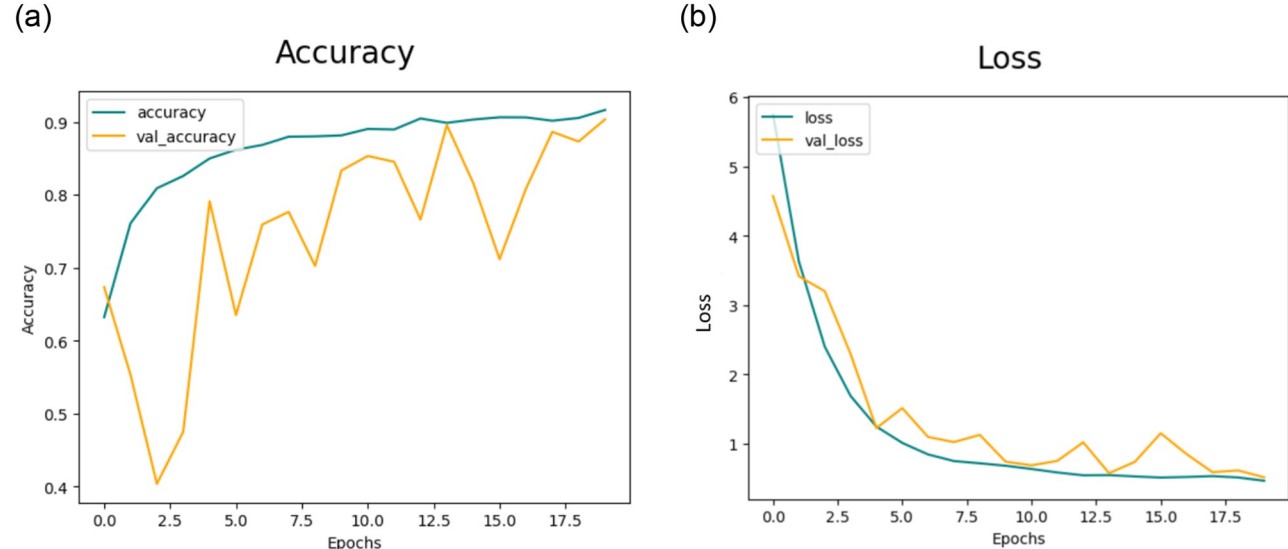

**Fig 14.** (a) Accuracy of the Model CNN-3 with Data Augmentation (b)Loss of the Model CNN-3 with Data Augmentation.

**Table 10. Class-specific statistics for the CNN-3 model with augmented data.**

| Classes | Accuracy % | Sensitivity % | Specificity % | Precision % | Recall % | F1 Score % | Support |
|---|---|---|---|---|---|---|---|
| DR | 93.90 | 76.36 | 98.49 | 92.96 | 72.98 | 81.74 | 470 |
| MH | 95.51 | 99.26 | 79.03 | 59.65 | 98.13 | 74.08 | 481 |
| ODC | 96.20 | 86.12 | 98.02 | 93.65 | 84.16 | 88.69 | 473 |
| WNL | 94.50 | 61.96 | 98.97 | 95.05 | 61.94 | 74.98 | 465 |

retinal diseases from the healthy group, we compare its performance with relevant studies in the literature. In this portion, we examine the outcomes of our suggested method in comparison to previous work. After acquiring the data we carried out preprocessing, which was utilizing data augmentation to enlarge the image dataset. This helps us to train our model accurately and reduce the effect of overfitting. The unwanted area is removed through cropping. Cropping helps to focus on the relevant part of the image (ROI) where the features of interest are located. In our dataset the images are in different sizes like 512 x 512 x 3, 2144 x 1424 x 3, and 4288 x 2848 x3 we resize all images to size 224 x 224 x 3. Next in preprocessing, we use a one-hot encoder because encoding is used in CNN models for data classification to transform categorical variables into a numerical format that can be easily processed by the network, preserving the relationships between categories and enabling the model to effectively learn from categorical data. After preprocessing we use CNN model for feature Extraction. The detailed information layer used in CNN is given in Table 3. The accuracy observed by CNN-1 for testing was around 88.72%, which is not particularly bad. The accuracy obtained from CNN-3 is also good but the model shows overfitting as shown in Fig 14. The comparison with all models that are implemented in this article with their training time and Testing accuracy is given in Table 11.

To classify retinal diseases, [27–29] used Ensemble learning and achieved 79.2% accuracy, 94.32% F1 score, and measured sensitivity ranging from 0.00-1.00 respectively. Authors [31, 32] use Deep learning algorithms to classify retinal diseases. The authors of [31] achieved

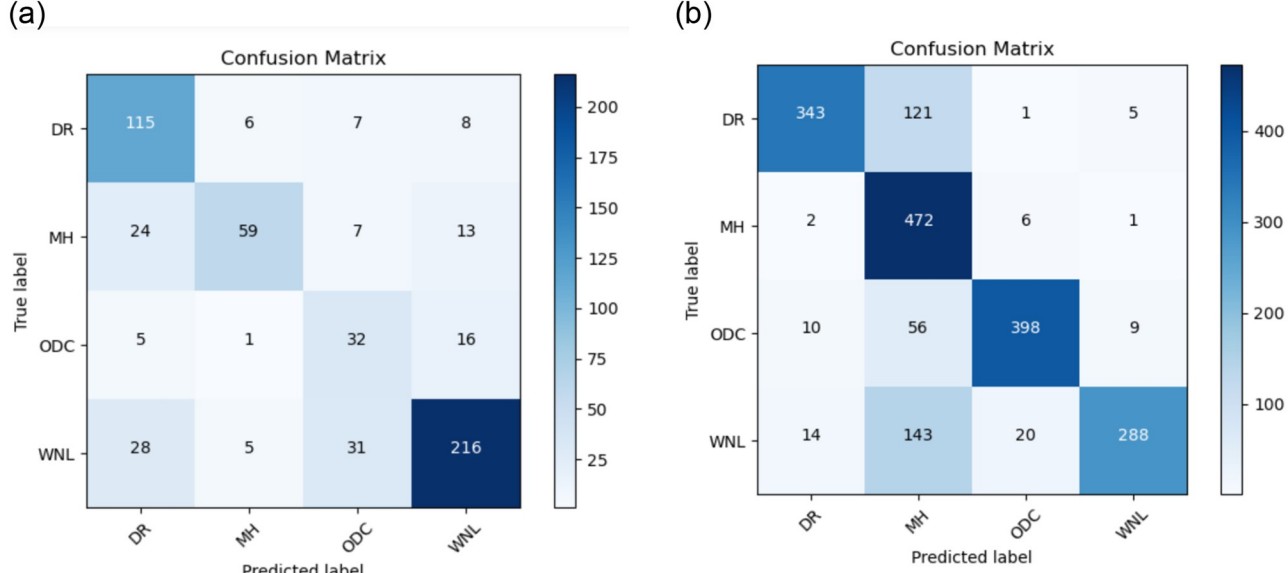

**Fig 15.** (a) Confusion Matrix for CNN-3 without the use of Data Augmentation (b) Confusion Matrix for CNN-3 with the use of Data Augmentation.

**Table 11. Overall performance metrics for CNN model across all classes.**

| Model | Training Time | Accuracy (%) |
|---|---|---|
| CNN-1 without the use of data augmentation | 1h 3min 44s | 73.12 |
| CNN-1 with the use of data augmentation | 2h 5min 51s | 88.72 |
| CNN-2 without the use of data augmentation | 3h 53min 54s | 69.81 |
| CNN-2 with the use of data augmentation | 10h 26min 18s | 84.86 |
| CNN-3 without the use of data augmentation | 27 min 11s | 72.95 |
| CNN-3 with the use of data augmentation | 1h 40min 16s | 90.47 |

88.4% for RVO, 85.2%, 93.8% for CSR, and 86.2% for Healthy. The authors of [32] achieved 89.10% for the RFMiD Dataset. [34] used different models for classification highest accuracy reached is 89.17 using ResNet152. [12] introduced a method EyeDeep-Net method consisting of CNN and achieved accuracy 76.04% testing accuracy for disease DR, MH, ODC, and Healthy class. In conclusion, we elaborate on the statistics of our proposed methodology. Our approach has notably enhanced the results. Author [12] used the Eye-DeepNet model to classify DR, MH, ODC, and WNL. Their accuracy was 82% for validation and 76.04% for testing. The result is provided in the Table 12.

This article provides the following information:

- **Three architectures based on deep learning:** We introduce and assess three distinct DL architectures tailored for the early identification of multiple retinal diseases, to enable timely intervention to prevent vision loss [39].

**Table 12. Comparative study of the proposed approach with previous work.**

| Reference | Year | No. of Classes | Model | Results |
|---|---|---|---|---|
| [28] | 2021 | 29 | Ensemble CNN | F1 Score 94.32% |
| [29] | 2022 | 29 | Ensemble Learning | Sensitivity for all 29 Classes Ranging from 0.00-1.00 |
| [31] | 2022 | 4 | Deep Learning | RVO 88.4% |
| | | | | DR 85.2% |
| | | | | CSR 93.8% |
| | | | | Healthy 86.2% |
| [12] | 2023 | 4 | EyeDeep-Net | Accuracy: |
| | | | | Validation 82.13% |
| | | | | Testing 76.04% |
| [27] | 2023 | 4 | Convolutional Ensemble | Accuracy 79.2% |
| [32] | 2024 | 5 | Deep Learning | Accuracy 89.10% |
| [34] | 2024 | 2 | ResNet152 | ResNet152 89.17%' |
| | | | Vision Transformer | Transformer 87.26% |
| | | | InceptionResNetV2 | Inc.ResNetV2 88.11% |
| | | | RegNet | RegNet 88.54% |
| | | | ConVNext | ConVNext 89.08% |
| **Proposed** | 2024 | 4 | CNN | MH 93.17% |
| | | | | Overall Acc. 89.81% |
| | | | | DR 91.95%, |
| | | | | ODC 94.60% |
| | | | | WNL 92.43% |

- **Data augmentation:** Our research highlights the importance of incorporating data augmentation techniques to boost model performance. For this specific context, leveraging data augmentation is an effective strategy to improve generalizability.

- **Comparative Evaluation with Existing Studies:** Our study incorporates a comparative analysis of prior research, with a specific emphasis on the RFMiD dataset. Compared to previous studies, our research provides a more accurate method of detecting retinal diseases.

By employing data augmentation techniques, we enhance the diversity of the training data. This not only helps in improving the model's robustness but also addresses the issue of overfitting, which is critical in medical imaging where obtaining large datasets can be challenging. Training and comparing multiple CNN models on the same dataset provides insights into which architecture is most effective for retinal disease classification. Including multiple classes three disease classes and one healthy class makes the model more versatile and clinically relevant. This multi-class approach mimics real-world diagnostic scenarios better than binary classification. This comparative analysis helps identify the strengths and weaknesses of each model. Providing a detailed analysis of the model performance e.g., accuracy, sensitivity, specificity, and precision for each class gives a comprehensive understanding of its diagnostic capabilities. This helps identify areas where the model performs well or needs improvement. By training and fine-tuning CNN models specifically for the classification of retinal diseases, we work to achieve higher classification accuracy compared to existing methods.

## 6 Conclusion

The classification of eye diseases is valuable for assessing the current health status of the eye, evaluating treatment outcomes, and selecting appropriate therapies. To facilitate early-stage identification and screening for eye disease patients, the development of a fully automated system is crucial. Such a system should be non-invasive, clinically reliable, reproducible, and have a manageable decision-making process. DL techniques combined with medical imaging offer a promising approach for providing detailed descriptions of detected diseases. Deep neural networks can learn hierarchical representations of images to aid in the diagnosis of various eye conditions. However, it is challenging to diagnose several eye conditions using a single neural network due to the similar appearance of fundus images of different diseases. To tackle this problem, this research suggests a DL-based CNN architecture. The objective of this model is to classify fundus images and provide a non-invasive detection for several vision disorders. The outcome of the suggested model is measured in terms of validation and testing accuracies, which are 89.81% and 88.72%, respectively. In the future, the model could also be used for other diseases and also in other medical fields. Image enhancement techniques and segmentation may also be applied for more accurate results. Future research may also extend the work to multi-label classification as datasets grow and model capabilities advance. Dataset deficiency is one of the major limitations in the medical field.

## Author Contributions

**Conceptualization:** Sara Ejaz, Mrim M. Alnfiai, Mona Mohammed Alnahari, Reemiah Muneer Alotaibi.

**Data curation:** Reemiah Muneer Alotaibi.

**Formal analysis:** Zeeshan Ashraf.

**Funding acquisition:** Mrim M. Alnfiai.

**Investigation:** Sara Ejaz, Raheel Baig.

**Methodology:** Sara Ejaz.

**Project administration:** Zeeshan Ashraf.

**Resources:** Mrim M. Alnfiai.

**Software:** Raheel Baig.

**Supervision:** Raheel Baig, Zeeshan Ashraf.

**Validation:** Raheel Baig, Mona Mohammed Alnahari, Reemiah Muneer Alotaibi.

**Visualization:** Mrim M. Alnfiai, Mona Mohammed Alnahari, Reemiah Muneer Alotaibi.

**Writing – original draft:** Sara Ejaz.

**Writing – review & editing:** Zeeshan Ashraf.

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
