## [Decision Letter · Decision Letter 0]

11 Jun 2024

PONE-D-24-19529A Deep Learning Framework for the Early Detection of Multi-Retinal DiseasesPLOS ONE

Dear Dr. Ashraf,

Thank you for submitting your manuscript to PLOS ONE. After careful consideration, we feel that it has merit but does not fully meet PLOS ONE’s publication criteria as it currently stands. Therefore, we invite you to submit a revised version of the manuscript that addresses the points raised during the review process.

We look forward to receiving your revised manuscript.

Kind regards,

Muhammad Mateen

Academic Editor

PLOS ONE

Journal Requirements:

"This research was funded by Taif University, Saudi Arabia, Project No. (TU-DSPP-2024-41)."

"The authors extend their appreciation to Taif University, Saudi Arabia, for supporting

this work through project number (TU-DSPP-2024-41)."

"This research was funded by Taif University, Saudi Arabia, Project No. (TU-DSPP-2024-41)."

5. Please provide a complete Data Availability Statement in the submission form, ensuring you include all necessary access information or a reason for why you are unable to make your data freely accessible. If your research concerns only data provided within your submission, please write "All data are in the manuscript and/or supporting information files" as your Data Availability Statement.

6. We note that Figure 2 in your submission contain copyrighted images. All PLOS content is published under the Creative Commons Attribution License (CC BY 4.0), which means that the manuscript, images, and Supporting Information files will be freely available online, and any third party is permitted to access, download, copy, distribute, and use these materials in any way, even commercially, with proper attribution. For more information, see our copyright guidelines: http://journals.plos.org/plosone/s/licenses-and-copyright.

Reviewers' comments:

Reviewer's Responses to Questions

**Comments to the Author**

1. Is the manuscript technically sound, and do the data support the conclusions?

Reviewer #1: Partly

Reviewer #2: Yes

2. Has the statistical analysis been performed appropriately and rigorously? 

Reviewer #1: Yes

Reviewer #2: No

3. Have the authors made all data underlying the findings in their manuscript fully available?

Reviewer #1: Yes

Reviewer #2: No

4. Is the manuscript presented in an intelligible fashion and written in standard English?

Reviewer #1: Yes

Reviewer #2: Yes

5. Review Comments to the Author

Reviewer #1: The aim of this study is to classify retinal images into for categories MH, ODC, DR, and healthy images using convolution neural networks.

My observations are summarized as follows:

1. It is not clear what gap your study has addressed by comparison with other retinal classification methods: apart from augmentation, what do the behavioral features represent(lines 464-468)?

2. A discussion of the reasons for choosing these CNN architectures (12 layers, 15 layers, 20 layers) is needed

3. Regarding the processing of the data set, the authors stated that "Each image is assigned to a single disease class, rather than having multiple labels." So in the case of retinal images, one image could have multiple diseases. Why didn't you consider a multi-label (multi-output) classification problem then?

4. It is not clear how transfer learning was used? What is the pre-trained architecture used?

5. How did you ensure independence between the training and testing datasets (eg, an original image can be used for training and its augmented variants can be used for testing)? Can the results be influenced in this case?

Reviewer #2: Dear authors,

It was a pleasure to read your manuscript titled "A Deep Learning Framework for the Early Detection of Multi-Retinal Diseases". The problem is formulated nicely and the proposed method addresses the problem accurately. Also, technical details such as data augmentation techniques used and network architecture are described in detail. The organization of the paper is also good. There are just some minor issues to be addressed:

1. Minor grammar mistakes. Between lines 366-373, two sentences are repeated. The manuscript is mostly free of grammatical errors. However, minor issues such as subject-verb agreement and punctuation should be reviewed.

2. Regularization techniques missing. This is the reason why your network is sometimes overfitting and why your accuracy curves are not smooth (and have a zig-zag pattern instead). Batch normalization can be added to the network.

3. Limited comparison with state-of-the-art. Although there is a comparison with some existing works, the study could benefit from a more extensive comparison with state-of-the-art models in retinal disease detection. This would provide a clearer context for the contribution and significance of the proposed framework.

In conclusion, it is a nicely conducted work which deserves to be published. It just needs some improvements in the above-mentioned points which are mostly related to the presentation of the work and highlighting its contributions.

6. PLOS authors have the option to publish the peer review history of their article (what does this mean?). If published, this will include your full peer review and any attached files.

Reviewer #1: No

Reviewer #2: No

---

## [Author Response · Author response to Decision Letter 0]

28 Jun 2024

Rebuttal Letter

Date: June 27, 2024

To: Muhammad Mateen (Academic Editor)

From: Dr. Zeeshan Ashraf (Corresponding Author)

Subject: Revised submission of the manuscript with comments incorporated

Manuscript Code Number: PONE-D-24-19529

Manuscript Title: A Deep Learning Framework for the Early Detection of Multi-Retinal Diseases

Dear Muhammad Mateen,

Thank you for inviting us to submit a revised draft of our manuscript entitled “A Deep Learning Framework for the Early Detection of Multi-Retinal Diseases” to your journal. We also appreciate the time and efforts you and each reviewer have dedicated to providing insightful feedback on ways to strengthen our research paper. Thus, it is with great pleasure that we resubmit our article for further consideration. We have incorporated changes that reflect the detailed suggestions you have graciously provided. We hope our modifications and the responses satisfactorily tackle all the issues and suggestions the esteemed reviewers have provided.

To facilitate the reviewer, the following is a point-by-point response to the questions and comments in your letter dated Wednesday, 12 June 2024.

Journal Requirements:

The manuscript meets PLOS ONE’s style requirements.

2. Please note that PLOS ONE has specific guidelines on code sharing for submissions in which author-generated code underpins the findings in the manuscript.

Yes. Code is available and shared without restrictions.

3. Thank you for stating the following financial disclosure. Please include this amended Role of Funder statement in your cover letter; we will change the online submission form on your behalf.

The role of the authors has been added to the manuscript and mentioned in the cover letter.

4. Thank you for stating the following in the Acknowledgments Section of your manuscript: We note that you have provided funding information that is not currently declared in your Funding Statement. However, funding information should not appear in the Acknowledgments section or other areas of your manuscript. We will only publish funding information present in the Funding Statement section of the online submission form.

Acknowledgments have been revised in the manuscript and mentioned in the cover letter.

5. Please provide a complete Data Availability Statement in the submission form, ensuring you include all necessary access information or a reason for why you are unable to make your data freely accessible. If your research concerns only data provided within your submission, please write "All data are in the manuscript and/or supporting information files" as your Data Availability Statement.

The Data Availability Statement has been included in the manuscript.

6. We note that Figure 2 in your submission contains copyrighted images. All PLOS content is published under the Creative Commons Attribution License (CC BY 4.0), which means that the manuscript, images, and Supporting Information files will be freely available online, and any third party is permitted to access, download, copy, distribute, and use these materials in any way, even commercially, with proper attribution. We require you to either (1) present written permission from the copyright holder to publish these figures specifically under the CC BY 4.0 license, or (2) remove the figures from your submission:

Figure 2 has been removed from the manuscript.

Reviewers' comments:

Reviewer #1: 

1. It is not clear what gap your study has addressed by comparison with other retinal classification methods: apart from augmentation, what do the behavioral features represent (lines 464-468)?

Thank you for mentioning this point. The research gap that this article has covered is mentioned in lines 501-515 on page number 20/24. The comparison with other methodologies in Literature is updated in Table 12 in page number 19/24. 

2. A discussion of the reasons for choosing these CNN architectures (12 layers, 15 layers, 20 layers) is needed.

Thank you for highlighting this point. The reasons for choosing these CNN architectures (12 layers, 15 layers, 20 layers) is mention in lines 295-303 in page number 9/24.

3. Regarding the processing of the data set, the authors stated that "Each image is assigned to a single disease class, rather than having multiple labels." So in the case of retinal images, one image could have multiple diseases. Why didn't you consider a multi-label (multi-output) classification problem then?

Thank you for addressing this point. The reason for choosing the single label image is mentioned in lines 215-223 in page number 6/24 and reason for multi-label (multi-output) classification problem is mentioned in line 532-535 in page number 20/24.

4. It is not clear how transfer learning was used? What is the pre-trained architecture used?

Thank you for pointing this issue. It was a draft mistake as transfer learning is not used in this article. That’s way removed this section. 

5. How did you ensure independence between the training and testing datasets (eg, an original image can be used for training and its augmented variants can be used for testing)? Can the results be influenced in this case?

Thank you for suggesting this idea. While using the augmentation technique original as well as augmented data is used for training and testing as mention in lines 203-210 in page number 6/24. The experiment is also performed using original data for training and augmented data for testing. When Experiment was performed 471 images for Diabetic Retinopathy, 334 for Media Haze, 172 for Optic Disc Cupping and 931 for Healthy class were used for training. For testing, 20% of the original images from each class were used, along with the augmented data. The Model shows overfitting as shown in Graph given below.

(a) 

(b)

Reviewer #2: 

1. Minor grammar mistakes. Between lines 366-373, two sentences are repeated. The manuscript is mostly free of grammatical errors. However, minor issues such as subject-verb agreement and punctuation should be reviewed.

Thank you for reading the manuscript thoroughly. The repetition error has been resolved. Proofreading has resolved the subject-verb issue.

2. Regularization techniques missing. This is the reason why your network is sometimes overfitting and why your accuracy curves are not smooth (and have a zig-zag pattern instead). Batch normalization can be added to the network.

Thank you for mentioning this issue. Now, experiments are performed by adding regularization technique L2 Regularization, increasing the Dropout Rate from 40% to 50%, and added Data Augmentation to reduce overfitting as mentioned in Tables 2 and 4 (in manuscript page number 8/24 and 10/24). Graphs with augmented data are updated as shown in Figures 8, 11, and 14 (in manuscript in page number 14/24, 16/24, 17/24). In this scenario, Batch Normalization does not reduce the overfitting issue as shown in the graph for CNN-3 (20 Layer CNN) (in manuscript). Some other experiments are also performed with Batch Normalization to check its behavior for this scenario the model shows overfitting as given below.

(a) 

(b)

3. Limited comparison with state-of-the-art. Although there is a comparison with some existing works, the study could benefit from a more extensive comparison with state-of-the-art models in retinal disease detection. This would provide a clearer context for the contribution and significance of the proposed framework.

Thank you for assisting. The comparison Table 12 is updated and added data in lines 478-488 page number 19/24. 

Thank you very much to the editor and the reviewers for your precious suggestions！

Sincerely,

Dr. Zeeshan Ashraf

---

## [Editor Report · Decision Letter 1]

4 Jul 2024

A Deep Learning Framework for the Early Detection of Multi-Retinal Diseases

PONE-D-24-19529R1

Dear Dr. Ashraf,

We’re pleased to inform you that your manuscript has been judged scientifically suitable for publication and will be formally accepted for publication once it meets all outstanding technical requirements.

Kind regards,

Muhammad Mateen

Academic Editor

PLOS ONE

Additional Editor Comments (optional):

All the comments have been addressed.
---

## [Editor Report · Acceptance letter]

16 Jul 2024

PONE-D-24-19529R1 

PLOS ONE

Dear Dr. Ashraf, 

I'm pleased to inform you that your manuscript has been deemed suitable for publication in PLOS ONE. Congratulations! Your manuscript is now being handed over to our production team.

Kind regards, 

on behalf of

Dr. Muhammad Mateen 

Academic Editor

PLOS ONE